# Synthesis of a Polymer Composite Based on a Modified Aminohumic Acid Tuned to a Sorbed Copper Ion

**DOI:** 10.3390/polym15061346

**Published:** 2023-03-08

**Authors:** Alma Khasenovna Zhakina, Zainulla Muldakhmetov, Tatyana Sergeevna Zhivotova, Bibigul Bagdatovna Rakhimova, Yevgeniy Petrovich Vassilets, Oxana Vasilievna Arnt, Arailym Alzhankyzy, Almat Maulenuly Zhakin

**Affiliations:** 1Limited Liability Partnership Institute of Organic Synthesis and Coal Chemistry of the Republic of Kazakhstan, Karaganda 100008, Kazakhstan; 2Non-Commercial Joint Stock Company, Department of Biomedicine, Karaganda Medical University, Karaganda 100008, Kazakhstan

**Keywords:** composite, natural polymer, humic acids, template, tuning, sorption

## Abstract

A composite based on amino-containing humic acid with the immobilization of multi-walled carbon nanotubes preliminarily tuned to a copper ion has been obtained. The synthesis of a composite pre-tuned for sorption by the local arrangement of macromolecular regions was obtained by introducing multi-walled carbon nanotubes and a molecular template into the composition of humic acid, followed by copolycondensation with acrylic acid amide and formaldehyde. The template was removed from the polymer network by acid hydrolysis. As a result of this tuning, the macromolecules of the composite “remember” conformations that are favorable for sorption, i.e., adsorption centers are formed in the polymer network of the composite, capable of repeated, highly specific interaction with the template and the highly selective extraction of target molecules from the solution. The reaction was controlled by the added amine and by the content of oxygen-containing groups. The structure and composition of the resulting composite were proven by physicochemical methods. A study of the sorption properties of the composite showed that after acid hydrolysis, the capacity increased sharply compared to a similar composite without tuning and a composite before hydrolysis. The resulting composite can be used as a selective sorbent in the process of wastewater treatment.

## 1. Introduction

The pollution of reservoirs with toxic metals (TM) is an urgent environmental problem for many regions of the world [1,2,3]. The main sources of TM intake into natural waters are wastewater from various industries. Given the large volumes of wastewater and the low concentration of TM in them, the methods of their treatment should be effective, affordable and environmentally safe. Sorption methods meet these requirements to the greatest extent [4,5,6,7,8]. Due to their efficiency, cost-effectiveness and environmental safety, sorbents have become widely used technologically. Various materials, including composite materials, are offered as sorbents for the extraction of toxic metals [9,10]. It is worth noting a general pattern: the higher the concentration of surface functional groups (SFG) and the specific surface of the composite material, the higher its sorption capacity. Thus, composite materials with a high specific surface area and grafted SFG can act as highly effective sorbents, but the difficulties of separating the solid and liquid phases limit their practical application. Despite the obvious progress in this area over the past decade, the search for highly effective means and methods of purification of water resources contaminated with TM ions is relevant [11,12,13,14,15]. Recent studies by a number of authors have focused on the development of composite materials as sorbents that combine the “desired” properties (such as the presence of SFG, developed specific surface area, mechanical strength, chemical resistance, etc.) of each of the components [16,17,18,19,20]. One of the approaches is to use natural polymers, such as humic acids, valuable products of the chemical processing of coal waste, to obtain composite sorbents suitable for the sorption method.

Humic acids (HA) are a wide class of high-molecular compounds that differ in the structure, composition and content of molecular fragments. Depending on the metamorphism, genesis and degree of oxidation, both low-molecular and high-molecular compounds with different content of aromatic and aliphatic fragments and functional groups may predominate in the composition of HA. It is known that the number of functional groups in the structure of HA macromolecules is an important characteristic that determines their reactivity and physicochemical properties. The extraction of HA does not present great economic and technological difficulties [21,22]. As for the multifunctionality of HA, in this regard, the development of methods for obtaining composite material by the structural modification of HA with new functional fragments in its macromolecules, studying their composition and properties, is promising.

Recent studies have made it possible to identify amino-containing humic compounds as the most promising objects that allow the creation of composite materials with fundamentally new properties. When combining a natural polymer and amino-containing compound, it is possible to obtain unique composite materials [23,24].

Currently, the creation of polymer composites with molecular imprints due to their ability to recognize metal ions and the ability to selectively sorb certain ions from solutions has become one of the most important topics for scientific researchers. Research in this direction opens the way for the synthesis of new composite materials based on HA and expands the boundaries of knowledge about their structure, properties and applications. It should be noted that HA is increasingly being used as a natural sorbent for cleaning man-made environments of toxic metals. The presence of a variety of oxygen-containing functional groups in combination with aromatic, heterocyclic and other groupings ensures the ability of HA to enter into almost any type of interaction: ionic, redox, donor-acceptor and sorption interactions. Numerous studies have proven that HA can bind almost all types of ecotoxicants, including transition metal ions [25,26,27,28].

Previously, we [29] investigated methods for obtaining composites based on amino-containing HA. The introduction of amino-containing compounds into HA, which are more prone to the formation of donor–acceptor bonds, made it possible to increase both the complexing properties of HA and, at the same time, give them polyampholytic properties. However, the use of the obtained products in technological processes is complicated by changes in the molecular structure of sorbents under the influence of chemical factors. In this regard, it is of interest to increase the chemical stability and mechanical strength of nitrogen-containing HA derivatives by modifying them with other reagents, in order to obtain chemically resistant and mechanically strong products for the selective binding of the target metal.

## 2. Materials and Methods

### 2.1. Materials

As a raw material in the synthesis of the composite, HA isolated from the oxidized coals of the Shubarkol deposit by alkaline extraction with further precipitation with mineral acid was used. HA had the following characteristics, %: humidity (W^a^)—12.1, ash content (A^a^)—22.0, carbon (C^g^)—36.3, hydrogen (H^g^)—3.73, nitrogen (N^g^)—0.70, sulfur (S^g^)—1.02, oxygen (O^g^)—58.9, (Σ(COOH + OH))—5.0 mg-eq/g. Multi-walled carbon nanotubes of the Taunit brand (manufactured by Nanotechcenter LLC, Tambov, Russia) were used as a modifier. The activation of multi-walled carbon nanotubes (MWCNTs) was carried out according to the method developed earlier by us [30]. Acrylamide (A) was used as an amino-containing compound (C_3_H_5_NO, M = 71.08 g/mol, produced by Sigma-Aldrich, St. Louis, MO, USA). The crosslinking agent used was formaldehyde, (37% aqueous solution, d = 1.09 g/cm^3^, produced by Sigma-Aldrich); the molecular template (M) was CuSO_4_·5H_2_O (GOST 19347-2014, M = 249.68 g/mol). A solution of 1 N HCl was prepared from a standard titer (TU2642-001-33813273-97, produced by “Uralhiminvest” CJSC, Ufa, Russia).

### 2.2. Synthesis of a Composite Based on Amino-Containing Humic Acid, with Immobilization of Multi-Walled Carbon Nanotubes, Pre-Tuned to Copper Ion

The synthesis of a composite based on amino-containing humic acid, with the immobilization of MWCNTs, pre-tuned to a copper ion, was carried out according to the method developed by us earlier in [30]. The content of copper ions introduced during tuning was 4 mg-eq per gram of composite. Further, this composite will be called HA:MWCNTs:M:A.

### 2.3. Study of the Stability of the Composite HA:MWCNTs:M:A for Acid Hydrolysis

Acid hydrolysis of the composite HA:MWCNTs:M:A was carried out as follows: the suspension of the composite HA:MWCNTs:M:A was poured with 1 N HCl solution, heated to 50 °C and kept for 30 min. Then, the composite was filtered, repeatedly washed with distilled water to a neutral medium and dried to a constant mass. Further, the resulting composite will be called HA:MWCNTs:A. Composite composition (%): HA 66.58, MWCNTs 0.13, A 33.29.

### 2.4. Study of the Sorption Properties of Composites

To study the sorption capacity of the obtained composites, experiments on the static sorption of copper ions were carried out. Sorption purification processes were carried out in static mode at 22 °C, in a liquid module, with a ratio of sorbent:sorbate = 1:25 and stirring for 24 h. After reaching sorption equilibrium, the composite was separated from the filtrate and the residual concentration of copper ions in the filtrate was determined by atomic emission spectroscopy using an atomic emission spectrometer with inductively coupled plasma, iCAR6500. The sorption capacity of composites was estimated by the value of the static exchange capacity of SEC, mg/g, and the sorption value R, %.

### 2.5. Studying the Composition of Composites

Elemental analysis of the initial HA and the obtained composites for the content of carbon, hydrogen, nitrogen, sulfur and oxygen was performed on the Elementar Unicube elemental analyzer.

The content of oxygen-containing groups in HA and composites was determined by direct and reverse conductometric titration using laboratory conductometer Anion-4100. (Ufa, Russia). Graphs of the dependence of the electrical conductivity on the volume of added acid were plotted and the number of oxygen-containing functional groups was calculated from the equivalence points and the corresponding volumes. Measurements were carried out sequentially on three hitches, and the average value of three experiments was taken as the final value. The measurement error was ±0.2%.

The IR spectra of the obtained composites were taken on the FSM-1201 IR Fourier spectrometer in KBr tablets. The range of wavenumbers was 4000–400 cm^−1^, and the error in determining the wavenumbers did not exceed 2 cm^−1^. Mathematical processing was carried out using a program for curve approximation and data analysis: Fityk 1.3.1 [31].

X-ray phase analysis (XRPhA) of the obtained composites was carried out on a D8 ADVANCE ECHO diffractometer using radiation from an X-ray tube with a Cu anode and a graphite monochromator on a diffracted beam. Diffractograms were recorded in the range of angles 15–100° 2θ, step 0.02° 2θ. Bruker AXS DIFFRAC.EVA v.4.2 software and the international databases ICDDPDF-2 and COD were used to identify phases and study the crystal structure.

The thermal stability of the composites was studied by differential thermal analysis (DTA) using a synchronous thermogravimetric differential analyzer, the Perkin Elmer STA 6000, in the measurement range: temperature up to 900 °C in a nitrogen atmosphere, υ = 10°/min.

The surface morphology of the obtained composites was studied using a MIRA 3 scanning electron microscope (TESCAN, Czech Republic, Brno). The scanning electron microscope (SEM,) is equipped with a system of detectors that register various signals formed as a result of the interaction of an electron beam with the sample surface. The secondary electron detector allows one to obtain images with topographic contrast. Meanwhile, the X-ray energy-dispersive microanalysis system X-Act (Oxford Instruments) allows one to locally determine the elemental composition on the sample surface.

The porosity characteristics of HA and the obtained composites were determined by obtaining isotherms of the low-temperature sorption–desorption of nitrogen on the Sorbi-MS (META, Novosibirsk, Russia) measuring complex using the SorbiPrep device at a liquid nitrogen temperature of 77 K, high purity Grade A (99.99%), by a dynamic method in the nitrogen current. To determine the specific surface area of the adsorption isotherm in the region of relative nitrogen pressure, the data were processed using the Brunauer–Emmett–Teller (BET) method, and the pore distribution by radii was determined by processing the points of the desorption isotherm using the Barrett–Joyner–Halenda (BJH) method. The specific surface of mesopores was determined by the STSA method.

## 3. Results and Discussion

The synthesis of the composite HA:MWCNTs:M:A included three stages (Figure 1).

At the first stage, a pre-polymerization complex based on humic acid (HA) was obtained with the immobilization of MWCNTs and a molecular template (M). Due to the formation of a pre-polymerization complex (HA:MWCNTs:M), polymer molecules are arranged and fixed in a certain way around the template molecule. The immobilization of the MWCNTs into the pre-polymerization complex was carried out using ultrasonic dispersion. Ultrasound helps to increase and regulate the porous structure, changing the chemical nature of the surface. At the second stage, by copolycondensation of the pre-polymerization complex (HA:MWCNTs:M) with acrylic acid amide (A) and formaldehyde, the synthesis of a composite (HA:MWCNTs:M:A) was performed, preconfigured for sorption by the local location of macromolecule sites. The introduction of a nitrogen atom into the composition of the pre-polymerization complex, which is more prone to the formation of donor–acceptor bonds with metal ions compared to oxygen atoms, will increase both the complexing properties of the composite and at the same time give them polyampholytic properties. The mechanism of composite (HA:MWCNTs:M:A) formation consists in the interaction of the modified polymer and the sorbed ion under conditions wherein the links of macromolecules still have sufficient mobility, with subsequent fixation of the resulting conformations optimal for sorption, which, in turn, should lead to a significant improvement in the sorption characteristics of the composite. At the third stage, a molecular template (M) was removed from the polymer mesh by acid hydrolysis to form a composite (HA:MWCNTs:A). The reaction was controlled by the attached amine using an elemental analyzer and by the content of oxygen-containing groups determined by conductometric titration methods. The results of the study are presented in Table 1.

As can be seen from Table 1, with the introduction of MWCNTs and A into the composite, the carbon content increased by 8–10%, and nitrogen by 0.4–0.7%. There was also a change in the content of oxygen-containing groups in the composites before and after hydrolysis. The yield of the composite HA:MWCNTs:M:A was 80.00%, and that of composite HA:MWCNTs:A was 81.50%.

Based on the CHN analysis data (Table 1), the atomic fractions and atomic ratios of the elemental composition were calculated for the studied samples. Thus, for the studied samples, the atomic fractions of the elements are equal to HA (C—0.29, H—0.35, N—0.005, O—0.35), HA:MWCNTs:M:A (C—0.38, H—0.29, N—0.008, O—0.32), HA:MWCNTs:A (C—0.40, H—0.27, N—0.010, O—0.31).

Atomic ratios (Table 2) calculated based on the atomic fractions of H/C, O/C, N/C show the number of hydrogen, oxygen and nitrogen atoms per molecule (HA, HA:MWCNTs:M:A, HA:MWCNTs:A) per a carbon atom. The smaller these ratios, the greater the role played by carbon atoms in the construction of the molecular structure. By the ratio of each of these pairs, one can judge the relative branching of the side chains, the degree of oxidation and their role.

The H/C and O/C atomic ratios make it possible to estimate the content of unsaturated fragments and oxygen-containing functional groups in the structure of the studied samples. The ratio H/C < 1 indicates the predominance of aromatic fragments in the structure of the sample, and if this ratio is in the range 1.0 < H/C < 1.4, then the structure of the sample is predominantly aliphatic. In accordance with this provision and the data of Table 2, the structure of the studied HA contains mainly fragments with a linear structure, and fragments with an aromatic structure predominate in the structure of the composite samples, which indicates the immobilization of MWCNTs in the composition of HA and the occurrence of the copolycondensation process. It should also be noted that, unlike HA, the composites are characterized by lower content of oxygen-containing functional groups. The decrease in oxygen-containing functional groups in the structure of the composites indicates their interaction with copper cations, not only through the ion exchange mechanism, but also with the formation of complexes with various oxygen-containing groups associated with both the alkyl chain C_alk_-O and the aromatic chain C_ar_-O.

Figure 2 shows the IR spectra of the synthesized composites, which are characterized by the following peaks. The transmission spectrum is obtained, normalized to the absorption maximum in region 1595 cm^−1^, and then inverted to the absorption spectrum; then, it is fitted with Gaussian contours to compare the intensity and reveal offsets. The large number of various functional groups in humic compounds and sampling-dependent IR light dispersion do not allow us to rely on the registration of IR spectra with a high resolution. The peaks obtained as a result of the approximation belong to the following functional groups. A broad absorption band with a maximum in the region of 3000–3500 cm^–1^ can be attributed to the vibrations of OH groups bound by intermolecular hydrogen bonds [32]. The decrease in the intensity of the C = O (carboxyl) stretching vibrations (1710 cm^−1^) indicates the occurrence of ion exchange during the sorption of Cu^2+^ cations by HA [33]. The appearance of a “shoulder” at the C = O absorption maximum of skeletal vibrations (at 1600–1640 cm^–1^) indicates the inclusion of reaction products of the −C = N group in the molecular structure [34,35]. Absorption at 1378 cm^−1^ is caused by deformation vibrations in the structure of N–H bond composites. Peaks at 1287 and 1256 cm^−1^ are associated with the C-O groups of carboxylic acids, esters and O-H groups of phenols; in the same region, they can show the noticeable absorption of the N-H group in various positions. In the spectra of the composite (HA:MWCNTs:A), the peak is shifted by 30 cm^−1^, which indicates the breakdown of the Cu–O bonds. The pronounced peaks at 1035 and 1034 cm^−1^ can be caused by C–O stretching and OH bending vibrations in alcohol groups. Peaks at 540–915 cm^−1^ are caused by bending vibrations of the aromatic ring.

The X-ray phase analysis (XRPhA) method was used to determine the composition of the composites. According to the X-ray diffraction data of the composite (HA:MWCNTs:M:A) (Figure 3), in addition to the main components contained in HA, low-intensity reflexes characteristic of two CuO compounds have also been established (PDF-01-080-1916)—monoclinic, spatial syngony Cc(9) and Cu(OH)_2_—orthorhombic, spatial syngony Cmc21(36).

As can be seen from Figure 3, for the composite HA:MWCNTs:M:A, there are two copper-containing phases, CuO and Cu(OH)_2_, the content of which does not exceed 15% of the total volume. The area ratio for the two compounds is CuO:Cu(OH)_2_ = 2:1, which indicates the dominance of CuO in the composition of the studied composite. The average size of the crystallites is 25–28 nm. It should also be noted that for the composite HA:MWCNTs:M:A, there is a characteristic decrease in reflexes characteristic of HA, which may be due to reactions of interaction with A, the presence of which is also present in the sample in the form of weak reflexes in the region 2θ = 20–25°. The main difference in the composite HA:MWCNTs:A is the absence of crystalline phases of CuO and Cu(OH)2 oxide in its composition, which indicates the removal of copper from the tuned composite during acid hydrolysis (Figure 3). There is a change in the degree of structural ordering of the composite HA:MWCNTs:A and a decrease in the size of crystallites, which may be due to the effect of structural changes. The average size of the crystallites is 18–21 nm.

The results of the study on the thermal stability of the obtained composites are presented in Figure 4.

As can be seen from Figure 4, the TG curve of the initial HA showed a small endothermic effect and the first weight loss in the range up to 120 °C (10 wt.%), associated with the release of physically adsorbed water. In the temperature range above 250 °C, weight loss (up to 50 wt.%) is due to the destruction of the aliphatic components of peripheral HA fragments, as well as the course of the primary decomposition reactions of organic substances, which reach a maximum at 350 °C. At 400 °C, the process of the decarboxylation of HA practically ends. At temperatures up to 500 °C, thermal degradation processes take place in the “core” of the HA. In addition, a total weight loss of approximately 80 wt.% was estimated.

The thermal decomposition of composites HA:MWCNTs:M:A and HA:MWCNTs:A showed a low-temperature loss of the main mass up to 18.36 wt.% in the temperature range 358–433 °C. Composites in the range up to 120 °C also had a small endothermic effect associated with the destruction of crystalline hydrates. The stage of intense weight loss is observed from 170 °C. Intensive weight loss ends at 500 °C. The total mass loss of the samples is estimated at around 32 wt.%. An increase in temperature to 900 °C does not lead to a significant change in the mass of the tuned composite both before and after hydrolysis, which is explained by the completion of the active pyrolysis stage, since, at higher temperatures, it occurs in the mineral component. When comparing the obtained results, the following conclusion can be drawn: the greater the content of free ion-exchange (functional) centers, the higher its thermal stability. For the studied samples, the thermal stability of the composites is higher compared to the original HA.

Figure 5 shows micrographs of the surface obtained during electron microscopic studies of the HA:MWCNTs:M:A composite, and Figure 6 shows the HA:MWCNTs:A composite. Comparative analysis of the microphotograms indicates a difference in surface morphology. Figure 5 and Figure 6 show differences in the shapes of the molecules. The surface of the HA:MWCNTs:M:A composite is rough, uneven and layered; there are agglomerates ranging in size from 98 nm to several microns. Upon further magnification of the micrographs, it becomes clear that these agglomerates consist of elongated tubes approximately 100 nm thick and a micron long, which are part of the composite. At the same time, spherical molecules are present, as well as medium and large aggregates of molecules of various shapes. The layered structure of the composite is explained by the stepwise copolycondensation of the pre-polymerization complex with a nitrogen-containing compound. On the surface of the composite, areas were found that differed in their phase compositions. This can be explained by the fact that, in these areas, the molecules of the composite are stretched over the surface, forming a thin monomolecular layer due to the formation of cationic bridges between the composite and the surface. EDS map data confirm that the main components of the composite (HA:MWCNTs:M:A) are carbon and oxygen, and there are inclusions of silicon, nitrogen and copper. The surface of the composite (HA:MWCNTs:A, Figure 6) after hydrolysis also has a multilayer structure containing nanotubes. In particular, after acid hydrolysis, a clear change is observed in the amorphous phase of the hydrolyzed composite. As can be seen from Figure 6, acid hydrolysis cut the MWCNTs into short nanotubes. At higher magnification, one can notice quite significant amounts of short nanotubes, as well as gaps between individual elements, which seem to be quite deep, while the elements themselves seem to be smoothed and have, apparently, melted edges. The EDS map confirms the removal of copper ions and release of pores from the HA:MWCNTs:A composite. The absence of copper is also confirmed by the overall spectrum of the map.

The porosity characteristics of the obtained composites in comparison with HA are presented in Table 3.

As can be seen from the data in Table 3, on the surface of the synthesized composite HA:MWCNTs:M:A, there are pores with a diameter of 50–60 nm (up to 37% of the surface) and nanopores with a size of up to 15 nm (up to 13% of the surface). A comparative analysis of the specific surface area of the obtained composites in comparison with the original HA showed its increase. The specific surface area determined by the BET method is 39.0 and 44 m^2^/g; the specific surface of mesopores determined by the STSA method is 42.0 and 87.4 m^2^/g; the volume of pores with a radius less than 47.7 nm is 0.016 and 0.026 cm^3^/g for composites HA:MWCNTs:M:A and HA:MWCNTs:A accordingly. The effect of acid hydrolysis on the specific surface of the mesopores of the resulting composite can be traced from the data determined by the STSA method. Hydrolysis uncorks additional pores, increasing the specific surface area of the mesopores of the composite HA:MWCNTs:A by an order of magnitude. Given the shape of the curve of the low-temperature adsorption and desorption of nitrogen and the presence of a hysteresis loop, the isotherm belongs to the fourth type of isotherm according to the IUPAC classification, which indicates the mesoporous structure of the studied samples, and this is confirmed by the pore size distribution. The presence of pores with a diameter of <50 nm in the resulting composite is obviously associated with voids formed between the copper fragments during their packaging, which is clearly visible in the images recorded by scanning electron microscopy.

The sorption properties of the obtained composites tuned to copper are given in Table 4. For comparative analysis, a composite (HA:MWCNTs:A) was prepared simultaneously under identical conditions, but without tuning to copper ions.

The study of the sorption properties of the composites showed that after acid hydrolysis, the sorption capacity of the HA:MWCNTs:A composite increased by a factor of 2 in comparison with the similar composite before hydrolysis and with the HA:MWCNTs:A composite without tuning. The effect of improving the sorption properties of Cu^2+^ for the composite HA:MWCNTs:A is 3.9 mg/g; for the composite HA:MWCNTs:M:A, it is 1.8 mg/g; and for the composite HA:MWCNTs:A without tuning, it 3.1 mg/g. The experimental data on the sorption properties once again confirm the assumption that there are pores in the system that correspond to the ionic radius of the hydrolyzed metal and the efficiency of the composite HA:MWCNTs:A, selectively tuned to the sorbed copper ions.

## 4. Conclusions

Thus, using the molecular imprinting method, a composite based on HA was synthesized with the immobilization of the MWCNTs and a molecular template, followed by copolycondensation with acrylic acid amide and formaldehyde, pre-tuned to the sorbed copper ion. The immobilization of the MWCNTs and the template into the pre-polymerization complex was carried out using ultrasonic dispersion. Ultrasound helps to increase and regulate the porous structure, changing the chemical nature of the surface. The mechanism of composite (HA:MWCNTs:M:A) formation consists in the interaction of the modified polymer and the sorbed ion under conditions wherein the links of macromolecules still have sufficient mobility, with subsequent fixation of the resulting conformations optimal for sorption, which, in turn, should lead to a significant improvement in the sorption characteristics of the composite. The removal of the molecular template from the polymer mesh was carried out by acid hydrolysis. The reaction was controlled by the attached amine using an elemental analyzer and by the content of oxygen-containing groups determined by conductometric titration methods. The composition and structure of the obtained composites have been proven by physico-chemical analysis methods: elemental analysis, conductometric analysis, IR spectroscopy, XRPhA, TGA and electron microscopy. The study of the sorption properties of the composites showed that after acid hydrolysis, the sorption capacity of the tuned composite increased by two times compared to a similar composite without tuning and with a composite before hydrolysis. The results of the study suggest the formation of cavities (imprints) in the composite HA:MWCNTs:A, which are capable of interacting with the target template molecules and increase the capacity of the sorbent. The synthesized composite can act as a sorbent with directed sorption activity.

## Figures and Tables

**Figure 1 polymers-15-01346-f001:**
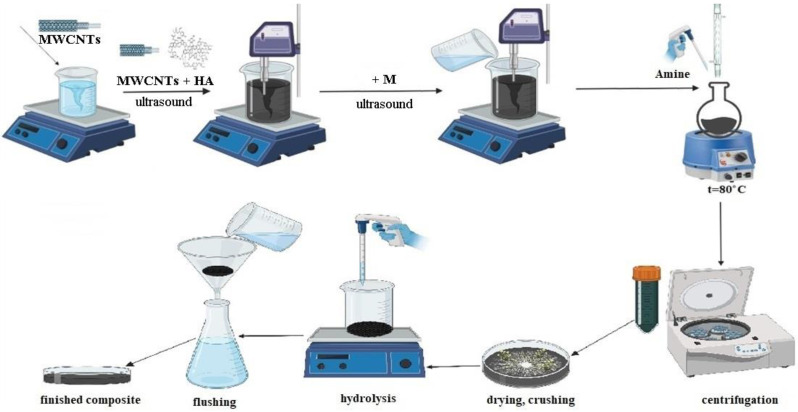
Scheme of synthesis of composites HA:MWCNTs:M:A and HA:MWCNTs:A (the scheme was created using a program available online: https://app.biorender.com/ (accessed on 2 December 2021)).

**Figure 2 polymers-15-01346-f002:**
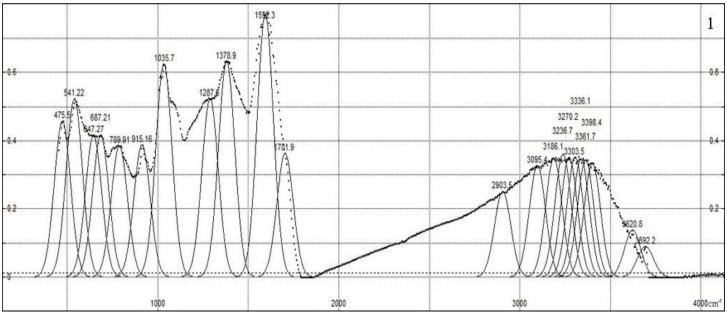
IR spectra of synthesized composites: 1—HA:MWCNTs:M:A, 2—HA:MWCNTs:A.

**Figure 3 polymers-15-01346-f003:**
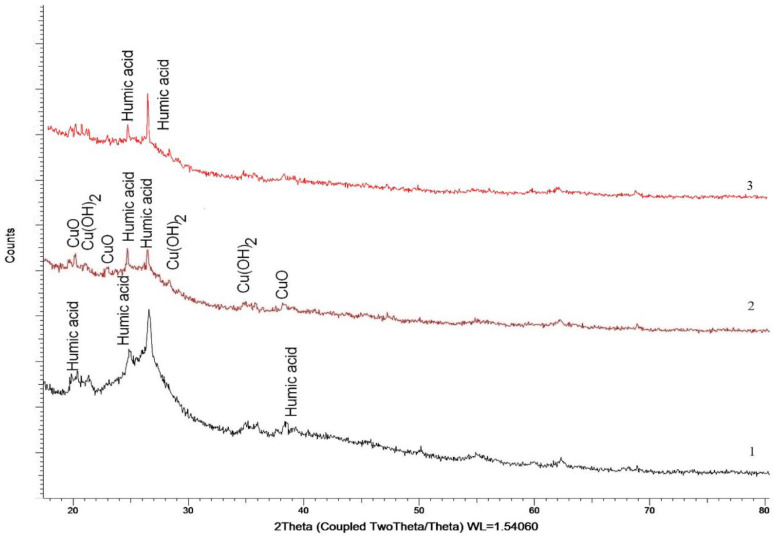
Diffractogram of the studied samples: 1—HA, 2—HA:MWCNTs:M:A, 3—HA:MWCNTs:A.

**Figure 4 polymers-15-01346-f004:**
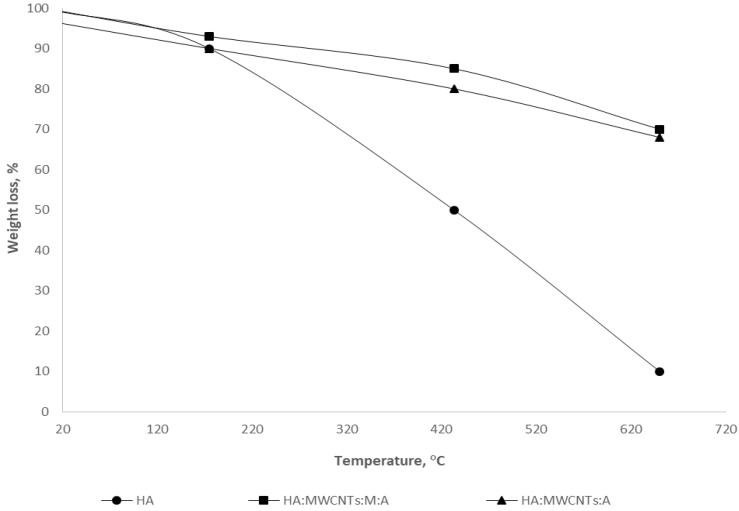
Thermal stability of HA and synthesized composites.

**Figure 5 polymers-15-01346-f005:**
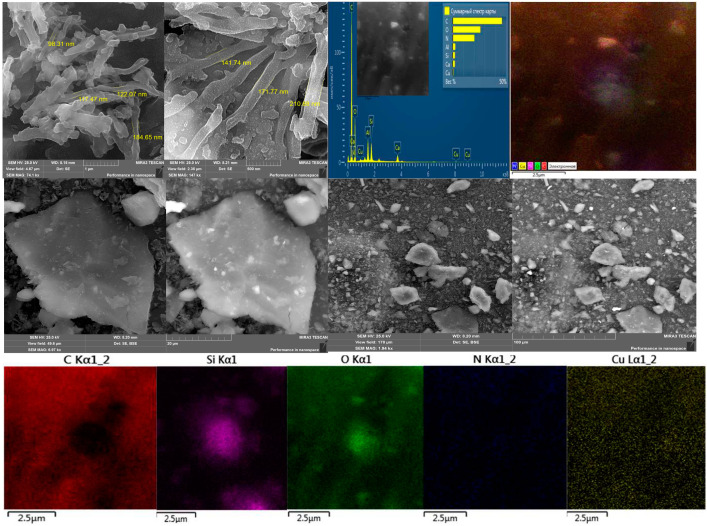
Topographic images of the surface, the EDS map and the total spectrum map of the composite HA:MWCNTs:M:A.

**Figure 6 polymers-15-01346-f006:**
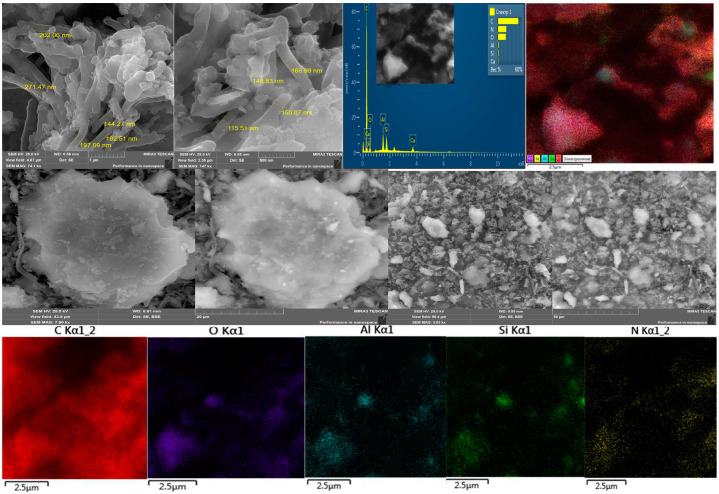
Topographic images of the surface, the EDS map and the total spectrum map of the composite HA:MWCNTs:A.

**Table 1 polymers-15-01346-t001:** Characteristics of synthesized composites.

Sample	C^g^, %	H^g^, %	N^g^, %	O^g^, %	Yield, %	Σ(COOH + OH), mg-eq/g
HA	36.30	3.73	0.70	58.25	75.01	5.0
HA:MWCNTs:M:A	44.58	2.83	1.13	50.42	80.00	4.6
HA:MWCNTs:A	46.74	2.66	1.39	48.44	81.50	4.8

**Table 2 polymers-15-01346-t002:** Atomic ratios of elements in the studied samples.

Sample	H/C	N/C	O/C
HA	1.21	0.02	1.21
HA:MWCNTs:M:A	0.76	0.02	0.84
HA:MWCNTs:A	0.68	0.03	0.78

**Table 3 polymers-15-01346-t003:** Results of determination of the specific surface of the composite before and after hydrolysis.

	HA	HA:MWCNTs:M:A	HA:MWCNTs:A
Specific surface area, m^2^/g (BET method)	14.1	39.0	44.0
Specific surface area of mesopores, m^2^/g (STSA method)	54.4	42.0	87.4
Total pore volume with R less than 47.7 nm, cm^3^/g	0.050	0.016	0.026
Distribution of pores relative to their total volume			
Pore diameter, nm	General content, %		
3.5	12.1	4.2	1.7
4.4	13.2	2.5	3.9
5.9	6.8	3.4	5.0
8.4	0.0	4.6	3.7
15.0	0.0	12.8	14.3
29.3	6.0	1.8	8.5
43.6	4.0	0.0	0.0
56.1	43.7	37.1	23.4
79.6	14.2	33.2	39.1

**Table 4 polymers-15-01346-t004:** Sorption capacity of composites.

Composite	Sorption	
	SEC, mg/g	R, %
HA:MWCNTs:M:A	1.8	38.50
HA:MWCNTs:A	3.9	83.41
HA:MWCNTs:A (without tuning)	3.1	65.40

## Data Availability

The data presented in this study are available on request from the corresponding author.

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
