# Peer review of "Synthesis of a Polymer Composite Based on a Modified Aminohumic Acid Tuned to a Sorbed Copper Ion"

_polymers, 2023, doi:10.3390/polym15061346_

Round 1

Reviewer 1 Report

1.     The quality of the Fig. 2 is very poor, please improved it. Also, give the main peaks from the functional groups in the figure.

2.     The paper mainly reports adsorption of metal pollutants, but there is no adsorbing rate;

3. “The appearance of a “shoulder” at the maximum absorption of C=O skeletal vibrations (at 1600-1640 cm-1) indicates the inclusion of reaction products of the group –C=N in the structure of molecules.” Pls add related refs, such as Inorganics, 10(2022) 202 and Micropor. Mesopor. Mat, 341(2022) 112098. A wide absorption band with a maximum in the region of 3000-3500 cm-1 can be attributed to vibrations of OH groups bound by intermolecular hydrogen bonds. This also could be cited, such as J. Solid State Chem. 318(2023) 123713. “Despite the obvious progress in this area over the past decade, the search for highly effective means and methods of purification of water resources contaminated with TM ions is relevant.” This sentences could not be separated as one part, also, it could be updated the current results, such as Mater. Today. Commum., 2022, 31,103514 and Coord. Chem. Rev. 2020, 406:213145

4. There is no adsorbing mechanism diagram.

5. Give SEM after the adsorbing Cu ions

Author Response

LOOK AT THE ATTACHMENT.

Reviewer 2 Report

Minor revision

- Abstract: precise the percentages of your choosen composit

- Figures can be improved to be more clear and adequate

- Cite tables and figures in the texte manuscript

- update your references by adding 2023

with regards

Author Response

LOOK AT THE ATTACHMENT.

Reviewer 3 Report

-          “Acrylic acid amide” – I am not sure about the compound. It is acrylamide?

-          L192. Please rephrase to be clear that the content increases with specific percent not from % to %.

-          Fig 2 – FTIR spectra are with a very low resolution. Also, the usance is to reverse the values in x- axes.

-          L 203 – the specified band are not visible in spectra. The same for many other bands. Please revise the whole part reffering FTIR

-          Some diffraction peaks in Fig 3 assigned to Cu containing phases can be also found in the HA diffractogram (about 20o). Please explain.

-          Figs 3 and 4 contains both HA – I think that the figures should be merged

-          I don’t understand how the thermal stability was determined, just in some marked points. Fig 5, Y-axes in the weight loss (not mass)

-          How it is possible that by including a soft material (acrylamide) to increase the thermal stability?

-          Figs 6 and 7– results from EDS are imposible to be understand. Also, why the differences in SEM images of the same sample? The presence of Cu in spectra is not clear.

-          Overall, the manuscript present results and less discussion (none about correlation between results) and I suggest to revise all manuscript and to improve the discussion part

Author Response

LOOK AT THE ATTACHMENT.

Round 2

Reviewer 1 Report

the authors have replied all my comments. accepted.

Author Response

Thank you for reviewing the article.

Reviewer 3 Report

Please find in the attached file, with red color, some comments directly related to authors answers

Author Response

We have made changes and additions to the manuscript based on your comments.
